# MicroRNAs: From Junk RNA to Life Regulators and Their Role in Cardiovascular Disease

Federica Amodio [1], Martina Caiazza [1], Fabio Fimiani [1], Paolo Calabrò [2,3] and Giuseppe Limongelli [1,4,*]

1 Inherited and Rare Cardiovascular Diseases, Department of Translational Medical Sciences, University of Campania "Luigi Vanvitelli", Monaldi Hospital, 80131 Naples, Italy; amodio.federica@yahoo.it (F.A.); martina.caiazza@yahoo.it (M.C.); fimianifabio@hotmail.it (F.F.)
2 Division of Cardiology, AORN Sant'Anna e San Sebastiano, 81100 Caserta, Italy; paolo.calabro@unicampania.it
3 Department of Translational Medical Sciences, University of Campania "Luigi Vanvitelli", 80131 Naples, Italy
4 Institute of Cardiovascular Sciences, University College of London and St. Bartholomew's Hospital, London WC1E 6BT, UK
* Correspondence: giuseppe.limongelli@unicampania.it; Tel.: +39-081-706-2815

**Abstract:** MicroRNAs (miRNAs) are single-stranded small non-coding RNA (18–25 nucleotides) that until a few years ago were considered junk RNA. In the last twenty years, they have acquired more importance thanks to the understanding of their influence on gene expression and their role as negative regulators at post-transcriptional level, influencing the stability of messenger RNA (mRNA). Approximately 5% of the genome encodes miRNAs which are responsible for regulating numerous signaling pathways, cellular processes and cell-to-cell communication. In the cardiovascular system, miRNAs control the functions of various cells, such as cardiomyocytes, endothelial cells, smooth muscle cells and fibroblasts, playing a role in physiological and pathological processes and seeming also related to variations in contractility and hereditary cardiomyopathies. They provide a new perspective on the pathophysiology of disorders such as hypertrophy, fibrosis, arrhythmia, inflammation and atherosclerosis. MiRNAs are differentially expressed in diseased tissue and can be released into the circulation and then detected. MiRNAs have become interesting for the development of new diagnostic and therapeutic tools for various diseases, including heart disease. In this review, the concept of miRNAs and their role in cardiomyopathies will be introduced, focusing on their potential as therapeutic and diagnostic targets (as biomarkers).

**Keywords:** miRNA; biomarkers; cardiomyopathy; hypertrophic cardiomyopathy; arrhythmogenic cardiomyopathy; dilated cardiomyopathy



## 1. Introduction

With age, the heart undergoes enormous stress and pathological stimuli that lead to cardiac remodeling and consequent cardiovascular disease [1,2]. During the last twenty years, non-coding microRNAs (miRNAs) have been identified as fundamental negative gene regulators. Among the functions regulated by these RNAs, there are also physiological and pathological aspects of the heart [3,4]. For a long time considered as junk RNA, today they are considered extremely important. MiRNAs control physiological functions such as the proliferation and differentiation of stem and progenitor cells, the function of cardiac myocytes, pacemaker cells, endothelial cells and smooth muscle cells [5]. These small sequences play a fundamental role in regulating cardiomyocyte contractility, maintaining heart rhythm, plaque formation, lipid metabolism and angiogenesis [6]. Even the stress response of the heart muscle is controlled by a precise spatiotemporal gene regulation of miRNAs [7,8].

MiRNAs play a fundamental role in the pathophysiology of myocardial remodeling, causing damage to cardiomyocytes, cardiac hypertrophy, cardiac fibrosis and abnormal inflammatory response, binding to multiple targets. The balancing and minute regulation

of different miRNAs are key to guiding cellular events towards functional recovery, and any variation can lead to detrimental effects on cardiac function following various insults. The discovery of miRNAs, therefore, changed our understanding of the regulation of gene expression.

As the expression pattern of miRNAs varies according to the type of heart disease, it is believed that dysregulation of miRNA expression can be detected in the blood of patients with cardiomyopathies, making them optimal candidates as non-invasive biomarkers for diagnosis, prognosis and therapeutic response [9–11]. The characteristic that makes miRNAs possible biomarkers and even potential therapeutic targets are their stability and presence in circulating biofluids [12]. Instead, their specificity concerning different cardiac diseases needs to be studied more.

In this review, we will analyze the main characteristics of miRNAs and their role in various cardiomyopathies such as hypertrophic cardiomyopathy, arrhythmogenic cardiomyopathy and dilated cardiomyopathy. It will be also mentioned the possible role that miRNAs could have as biomarkers of pathology, as well as an overview of future therapeutic approaches of miRNAs for cardiomyopathies.

## 2. MiRNAs: What Are They?

From an early age, we have always been taught the "Dogma of Biology", which states that DNA is transcribed into RNA and then translated into protein. This suggested that only coding RNA could determine gene expression. Only 1% of the human genome encodes genes which will then be translated into proteins [13]. The remaining 99% of the DNA was considered junk and therefore not useful for gene regulation. In recent years, the importance of non-coding RNAs (ncRNAs) as regulators of gene expression has finally been understood [14]. Although ncRNAs cannot encode proteins, an important role of ncRNAs in the regulation of gene expression has been observed, both in physiological and pathological conditions (Table 1) [15].

In this review, we will focus on a specific class of ncRNA, namely, microRNAs (miRNAs), which correspond to ~5% of the human genome [16,17]. MiRNAs are a class of single-stranded, non-coding RNA, approximately 18–25 nucleotides in length, which negatively regulate gene expression at the post-transcriptional level. They are highly conserved ncRNA and can be identified in animals, plants and viruses [18], and are thought to be a vital evolutionary component of gene regulation [19,20].

MiRNAs influence protein production by binding to messenger RNA (mRNA) via imperfect base pairing and complementary sequences in their target mRNA, resulting in translational repression or transcript degradation [21,22]. The mRNA and miRNA binding occurs through a sequence of 2–8 nucleotides [7] of the 5′ end of the miRNA with the 3′ untranslated region (3′UTR) of the target mRNAs [23–25].

It has been estimated that miRNAs control the activity of 30–50% of protein-coding genes [26]. Unlike transcriptional regulators, which have an activation and deactivation function in controlling gene expression, the various miRNA profiles appear to adapt the level of protein expression to changes in environmental conditions [6]. Due to the imperfect binding between the two strands, a single miRNA can influence the expression of many target genes in a cell and a target mRNA can be regulated by multiple miRNAs, as mRNAs host binding sequences for multiple miRNAs (Figure 1) [25,26]. MiRNAs are essential in various biological processes, including cell differentiation and proliferation, cell death and metabolism [27–30]. Dysregulation of miRNAs often disrupts critical cellular processes, leading to the onset and progression of various human diseases including heart diseases [31]. Furthermore, miRNAs can enter the circulation through microvesicles (exosomes) in an extremely stable form and can therefore be quantified through plasma sampling and then used as biomarkers [32,33].

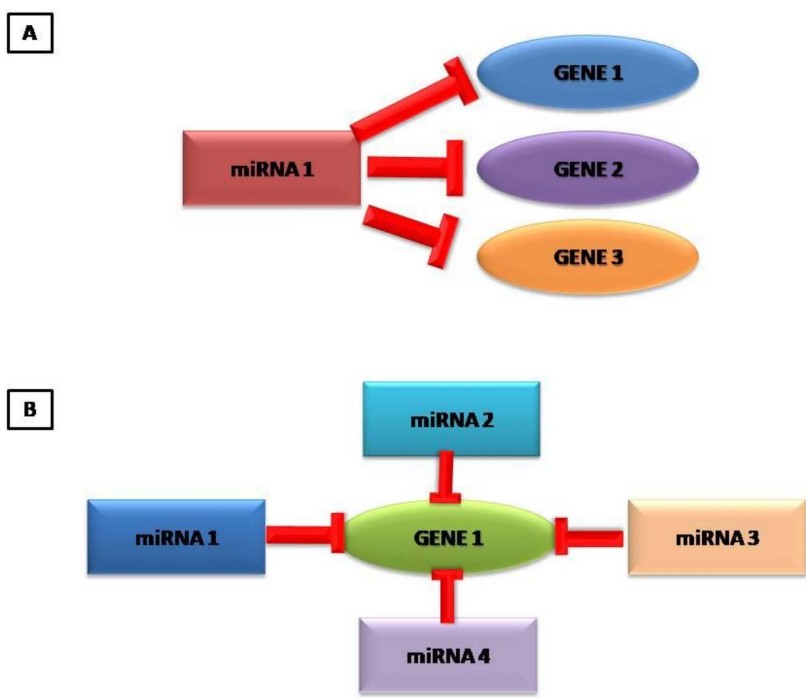

**Figure 1.** Graphical representation that a miRNA can control several genes (**A**) and that one gene can be controlled by different miRNAs (**B**).

**Table 1.** Brief description of the main ncRNAs.

| ncRNA Class | Function | References |
|---|---|---|
| Micro RNA (miRNA) | Fundamental role in gene regulation at the post-transcriptional level. They act either by cleaving the target mRNA or by inhibiting translation and therefore protein synthesis | [21] |
| long non-coding RNAs (lncRNAs) | They have various inhibitory functions by acting on transcription regulatory proteins (histone modifying enzymes and chromatin remodeling factors), mRNA and miRNA | [34,35] |
| circular RNAs (circRNAs) | Their function is not yet well known, however some experiments have shown that some of them can bind to specific proteins or to miRNAs blocking their functions. | [36,37] |
| Small nucleolar RNA (snoRNA) | Essential to drive nucleotide modifications and processing. | [38] |

### 2.1. The miRNA Factory: How They Work

MiRNAs are an evolutionarily conserved integral part of the cellular genome. Depending on the genomic location of the miRNA coding sequences, miRNAs can be transcribed as independent transcription units starting from intergenic regions or in the introns and exons of the protein-coding genes. Intergenic miRNAs are transcribed under the control of distinct promoters, while the transcription of intronic and exonic miRNAs is mainly controlled by their host gene promoters. The genes encoding miRNAs can be transcribed singly or in polycistronic clusters forming a long transcript which is then cleaved into the different miRNAs [22,39]. In the latter case, more than one miRNA is transcribed

together by the same transcription factor, generating a transcript with miRNAs belonging also to different families. Therefore, miRNAs from the same cluster can target multiple mRNAs [40]. Alternatively, miRNAs can be transcribed from introns of protein-coding genes, from introns or exons of non-coding genes, or even from 3′UTR of protein-coding genes [41].

The biosynthesis of miRNAs occurs in several enzymatic steps, both in the nucleus and in the cytoplasm (Figure 2) [23,42]. The miRNAs are transcribed into the nucleus by RNA polymerase II (RNA Pol II) into primary miRNAs (pri-miRNA), a long strand of hundreds to thousands of nucleotides that overlaps on itself with an imperfect pairing in a stem-loop structure with two flanking single-stranded regions at the end [43–45].

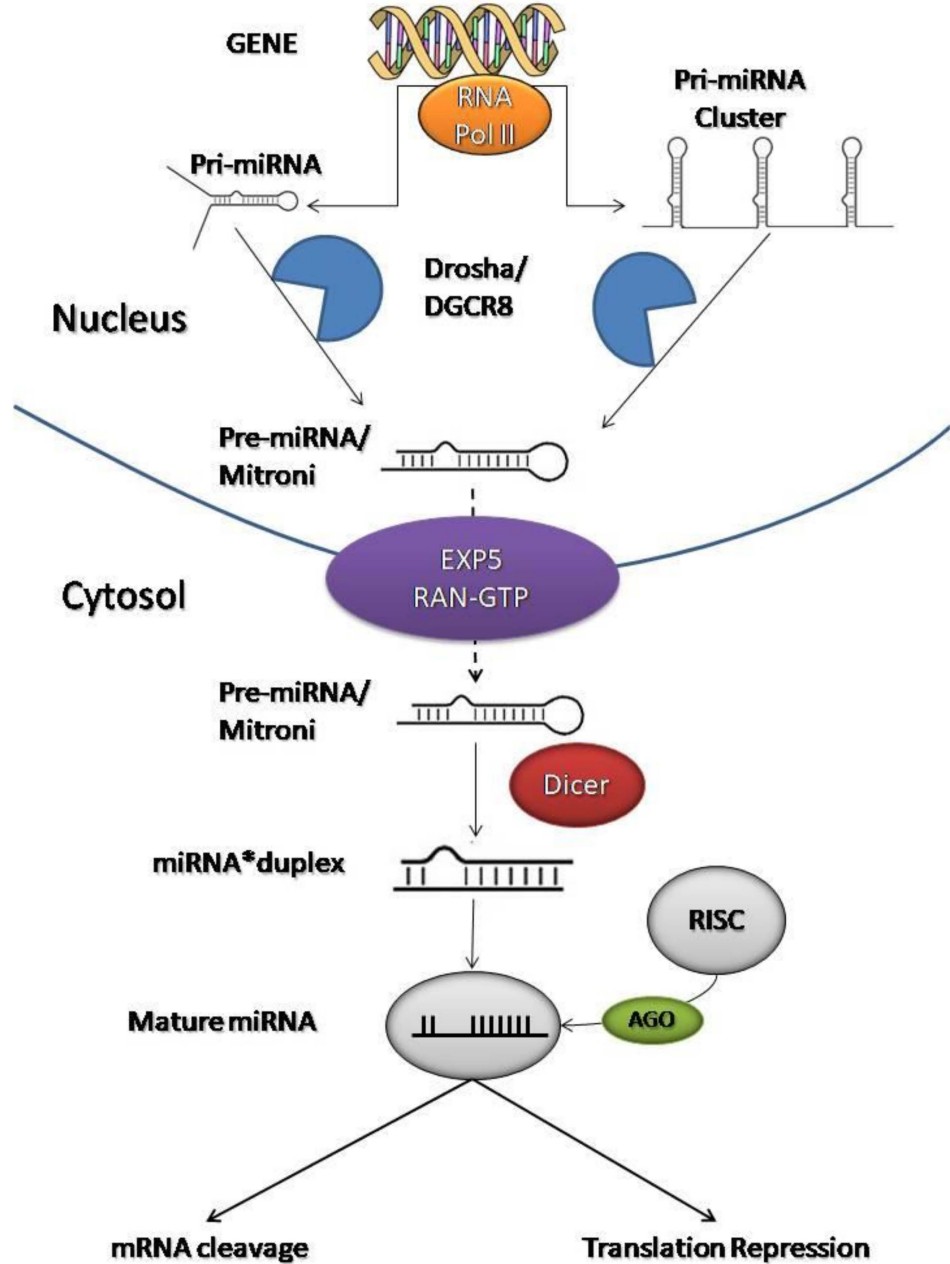

**Figure 2.** Graphical representation of miRNA biogenesis and how they inhibit mRNA translation at the post-transcriptional level.

At this point, it activates an enzyme called Drosha (protein complex containing the RNase III endonuclease) that is associated with a nuclear protein called DGCR8. Drosha

(with DGCR8) cleaves the 5′-end and 3′-end of the pri-miRNA and stabilizes the molecule, giving rise to precursor miRNAs (pre-miRNA) of ~70 nucleotides with a hairpin structure [22,39,44]. Then, the process continues in the cytoplasm.

Exportin 5 (EXP5) and RAN-GTP (GTP-dependent protein) form a transport machinery that recognizes a short sequence of 2–3 nucleotides at the end of the pre-miRNAs to transport them from the nucleus to the cytoplasm [22,46]. EXP5 also protects pre-miRNAs from the degradation process which avoids the accumulation of pre-miRNAs in the nucleus [47].

A final trimming step is performed in the cytoplasm by RNase III endonuclease called Dicer. Dicer interacts with the 5′-end and 3′-end of the hairpin and creates double-stranded miRNAs (miRNA*duplex) of approximately 22 nucleotides [48,49]. A double-strand presents an imperfect complementarity which will facilitate its division [50]. The miRNA*duplex splits to form two single strands: the guide strand, which acts as a functional unit, and the passenger strand [51,52]. The choice of the guide strand depends on the thermodynamic stability of the 5′-end of the miRNA*duplex [53]. In general, the strand with lower stability preferentially combines with Argonaute protein (AGO). This causes some miRNAs to have both of their strands loaded into the RNA-induced silencing complex (RISC) with the same proportion, while for others only one strand will dominate [53].

The guide strand combines with AGO in the RISC complex to become a mature and functional miRNA [54,55]. The passenger strand is often degraded or incorporated into microvesicles and released in the bloodstream [12,56]. Both mature miRNAs and pre-miRNAs can be found in microvesicles [12].

Thanks to the link with RISC, a mature miRNA is formed, ready to inhibit target mRNAs at the post-transcriptional level [22,57]. The RISC binding sites are complementary sequences to the 3′-untranslated region (3′UTR) of the mRNAs [58]. Therefore, this complex can direct mRNA towards miRNA through sequence complementarity. The miRNA sequence, incorporated in the RISC, binds to the mRNA by a seed sequence (first 2–8 nucleotides of the 5′ end of a miRNA).

When the miRNA sequence matches its target perfectly, AGO cleaves the mRNA resulting in direct degradation. More frequently, when complementarity is lacking, mRNAs translation is inhibited, blocking protein synthesis [59]. Therefore, AGO does not induce target cleavage, but represses translation by three different mechanisms: translation initiation block, elongation block, or deadenylation [60]. However, these processes will not be discussed in this review.

Thanks to the imperfect base-pairing with complementary sequences between miRNA and mRNA, a single miRNA can regulate multiple mRNAs targets which are involved in different biological processes and, conversely, a single mRNA can be regulated by several miRNAs.

Generally, gene suppression is partial, and one mRNA can have multiple binding sites for different miRNAs.

Not surprisingly, the binding of a single miRNA may not be sufficient to block the translation and several miRNAs are needed to regulate a single mRNA [18].

Mature miRNAs are indicated with the prefix "miR-" followed by an identification number that reflects their order of discovery. In case of miRNAs with sequences that differ only in one or two nucleotides, an additional letter or number is added to the name, e.g., "miR-120a" and "miR-120b" or "miR-232a-1" and "miR323a-2" [10,39].

Little is known about the half-life and degradation of mature miRNAs, which are stable in cells and have also been shown to play a central role in cell-to-cell communication [12,43]. Most miRNAs are located at the intracellular level, but some of them are released into the blood in association with proteins (e.g., Ago2, nucleophosmin 1 and HDL). They can be packaged in microvesicles, exosomes or apoptotic bodies in circulation, allowing resistance even to changes in temperature, pH and multiple freeze/thaw cycles.

They are present in all blood compartments, including plasma, platelets, erythrocytes and nucleated blood cells [57]. MiRNAs bind RNA-binding proteins or high-density

lipoproteins, forming miRNA–protein complexes which give them protection from circulating RNases [12].

*2.2. MiRNA as Biomarkers*

As previously mentioned, miRNAs are relatively stable within biofluids and tissues, allowing us to measure them reliably [61,62]. We indicated that mature miRNAs, pre-miRNAs and stand passengers can be found in the blood due to their inclusion in microvesicles. Therefore, it is possible to measure these molecules not only in the tissues, but also in the blood of patients with CMP. Several techniques are available to identify and quantify miRNAs, but the gold standard is the real-time PCR, which remains the most reliable technique. This technique can be used only with predefined primers to amplify and measure individual miRNAs in a sample. Alternatively, miRNA arrays could be used, which consists in miRNA hybridization on specific primers. MiRNA arrays allow obtaining a less accurate quantification than the previous one; however, they allow evaluating a greater number of miRNAs at the same time at lower costs.

Both of these techniques can be used only and exclusively for already known miRNAs, as they are based on predefined primer sequences. The identification of uncharacterized miRNAs can be evaluated with non-coding RNA sequencing technologies, essential for the identification of miRNAs with unknown sequences. In this way, it is possible to have a quantitative analysis of a complete miRNA transcriptome [63].

In recent years, advances in molecular technology, such as the Next-Generation Sequencing (NGS), have enabled the discovery of new disease sequences with higher yields and lower costs than conventional technologies [64]. NGS has become a fundamental tool for the clinical work-up of cardiomyopathies.

Therefore, these techniques can be used to evaluated miRNAs, which can play the role of pathological biomarkers. Not only is their expression important, but also the mutations in seed regions or the deletion of some miRNAs themselves can cause a pathology [65]. This is possible because an aberrant expression of miRNA profile during disease has been demonstrated. Not surprisingly, variations in miRNA expression have been found in cardiomyopathies, and specific plasma miRNAs have been identified in correlation with these illnesses. Different miRNA expression profiles have been studied in pathological and normal conditions and the result highlights that dysregulation of miRNA levels can detect a pathology.

The identification of miRNAs with the technique explained can allow evaluating the differential expression between healthy and diseased subjects [66,67]. In various studies, it has been shown that miRNA expression pattern was different and specific for different heart diseases; this led to the hypothesis that miRNAs could have etiological implications in the development of the disease. Therefore, their profile expression could be used as potential biomarkers for diagnosis, prognosis and response to therapy.

MiRNAs are present in target cells and tissues, in biological fluids, including plasma and serum, stored within microvesicles. The microvesicles confer protection from the cleavage of endonucleases and stability in biofluids and which makes them easily detectable in circulation [68–70].

With NGS technologies it is possible to obtain a rapid and accurate measurement of miRNAs which can be also measured with real-time PCR (even in small volumes of biofluids) [71,72]. The stability and diversity of miRNAs in combination with the available technologies make them new and promising diagnostic biomarkers [73,74]. However, miRNAs also have some limitations [75]. Unfortunately, miRNA detection is time-consuming and rapid diagnosis is not possible; there are no standardized laboratory protocols for the extraction, dosage and normalization of the analysis, so for now they may not be usable [76]. Furthermore, different miRNAs can regulate more genes and, at the same time, several genes can be regulated by more miRNAs, so there is a low correlation between miRNA and specific pathologies.

## 3. MicroRNA and Cardiomyopathies

In genetic diseases, pathogenesis mechanisms are often unknown or difficult to interpret. Mutations in a specific gene can be associated with different phenotypes. In this perspective, miRNAs and other ncRNA represent a further mechanism to be investigated in homogeneous groups of patients to better understand the mechanisms associated with particular pathologies.

MicroRNAs are a group of short, non-coding and endogenous RNAs that regulate gene expression through the sequence-specific recognition of their target transcripts [18]. They are involved in various cellular processes such as apoptosis, proliferation or differentiation. Aberrant levels of miRNA are implicated in numerous pathophysiological conditions, including cardiomyopathies. The overexpression or underexpression of certain miRNAs also plays a crucial role in cardiomyopathies such as hypertrophic cardiomyopathy (HCM), dilated cardiomyopathy (DCM) or arrhythmogenic cardiomyopathy (AC), demonstrating that several cardiomyopathies can have a specific miRNA signature [12].

MiRNAs are involved in cardiac biogenesis. Blocking specific genes essential for miRNA biogenesis, such as *DROSHA*, *DGCR8*, *AGO* or *Dicer*, in the heart tissue of murine embryos, results in an interruption of gestation due to severe developmental defects of the heart and blood vessels. However, the deletion of individual miRNAs is not lethal [6]. Thanks to several studies, we now know that heart muscle phenotypes are tightly regulated by multiple miRNA species [77]. Cardiomyopathies are a very heterogeneous group of diseases that affect the heart muscle. They are generally characterized by changes in the size of the heart chambers, ventricular wall thickness or contraction abnormalities.

The main cardiomyopathies affecting the population are HCM (1:500), DCM (1:2.500) and ACM (1: 5.000) [78,79] which we will deal with in this review. Table 2 shows the miRNAs studied in this review and their associated cardiomyopathies.

**Table 2.** Expression of the different miRNAs in human cardiomyopathies and the main target genes according to Targetscan (http://www.targetscan.org/vert_71/ accessed on: 17 October 21).

| MiRNA | Regulation | Cardiomyopathy | Reference | Gene Target |
|---|---|---|---|---|
| miR-590-5p | Upregulated | Hypertrophic | [80] | *TRIM, JPH1, POM121C* |
| miR-92a | Upregulated | Hypertrophic | [80] | *STAT, SUMO2, TBC1D1* |
| miR-483-5p | Upregulated | Hypertrophic | [81] | *APOL, DLL4, FHL2* |
| miR-29a | Upregulated | Hypertrophic | [82] | *PRTEN, AKT, NFAT, GSK3B, Elastin* |
| miR-133 | Downregulated | Hypertrophic | [83] | *CTGF, SERCA2a, NFATC4, MYH, SERCA* |
| miR-155 | Downregulated | Hypertrophic | [84] | *SOCS1, MEF2A, JARID2 AT1R* |
| miR-1 | Downregulated | Hypertrophic | [85] | *BCL1, CBX6, CCND1, CREB* |
| miR-204 | Upregulated | Hypertrophic | [86] | *ATXN1, CAPRIN1, CREB, OGT . . .* |
| miR-139-5p | Downregulated | Hypertrophic | [87] | *c-JUN, SRSF, Est-1, MEIS1, ZFX* |
| miR-20 | Upregulated | Hypertrophic | [88] | *STAT3, ATF2, DVL3* |
| miR-3135b | Upregulated | Dilated | [89] | *FLNC, PRX, RBL1* |
| miR-3908 | Upregulated | Dilated | [89] | *ADD2, FRMD4B, PDE11A* |
| miR-5571-5p | Upregulated | Dilated | [89] | *PPP2R2B, BMP7, TCF21, PSEN1* |
| miR-148a | Downregulated | Dilated | [90] | *gp130, AKT,ITPR2* |
| miR-185 | Upregulated | Dilated | [91] | *TFPI, Ctgf, ARHGEF, CAMK2* |
| miR-1251 | Upregulated | Arrhythmogenic | [92,93] | *TMEM, ANK1, PROX1* |

**Table 2.** *Cont.*

| MiRNA | Regulation | Cardiomyopathy | Reference | Gene Target |
|---|---|---|---|---|
| miR-21-3p, miR-21-5p | Upregulated | Arrhythmogenic | [92,93] | *PITX2, CADM1, PVRL3, SLMAP* |
| miR-212-3p | Upregulated | Arrhythmogenic | [92,93] | *PLXNA2, PRDM16, TCF, PKP4* |
| miR-34a-5p | Upregulated | Arrhythmogenic | [92,93] | *HCN, JPH, PKP2* |
| miR-135b | Upregulated | Arrhythmogenic | [92,93] | *ERBB, FOXO1,TMEM, SCN5A* |
| miR-138-5p | Downregulated | Arrhythmogenic | [92,93] | *WNT9A, BMPR, AKAP11* |
| miR-193-3p | Downregulated | Arrhythmogenic | [92,93] | *ALOX5, SOX2, L-MYC, KLF4* |
| miR-302 | Downregulated | Arrhythmogenic | [92,93] | *FMR1, CAMTA1* |
| miR-491-3p | Downregulated | Arrhythmogenic | [92,93] | *WNT, BMPR2, TGFBR2* |
| miR-575 | Downregulated | Arrhythmogenic | [92,93] | *EPB41L5, HCN1, HCN4* |
| miR-4254 | Downregulated | Arrhythmogenic | [92,93] | *CDR1AS, COL4A, HSPB7* |
| miR-4643 | Downregulated | Arrhythmogenic | [92,93] | *RBM20, RAC1, VCAM1, PTPRC* |
| miR-320a | Downregulated | Arrhythmogenic | [94] | *CDH2, CTNNA3, DSC2* |
| miR-144-3p | Upregulated | Arrhythmogenic | [95] | *CTNNA3, AREG, PROS1* |
| miR-145-5p | Upregulated | Arrhythmogenic | [95] | *CDH2, DAG1, CITED2, TLL1, PAK7* |
| miR-185-5p | Upregulated | Arrhythmogenic | [95] | *DLG2, NOX5,PRRT2* |
| miR-494 | Upregulated | Arrhythmogenic | [95,96] | *PTEN, ROCK1, CaMKIIδ, FGFR2, LIF* |

*3.1. Hypertrophic Cardiomyopathy*

HCM is the most common primary cardiomyopathy (1:500) with heterogeneous clinical and genetic characteristics, the leading cause of sudden cardiac death (SCD) in adolescents and athletes [97–99]. It is an autosomal dominant disease with incomplete penetrance.

HCM is a highly complex and heterogeneous disease as regards not only the number of associated mutations, but also the severity of the phenotype and the frequency of complications, such as heart failure and SCD [66,100]. HCM is also characterized by the varying degree of left ventricle hypertrophy (LVH) [101], symptoms and risk of SCD [102] or heart failure [103,104]. The disease is characterized by the so-called "myocardial disorder" which involved the hypertrophic non-dilated left ventricle. The pathological mutations generally affect genes that encode proteins belonging to the contractile components of the heart. Thus, mutations can occur in one or more of the eight genes encoding sarcomere proteins [105,106]. Seventy percent of HCM-causing mutations affect cardiac myosin binding protein C (*MYBPC3*) genes with nonsense mutations, as well as cardiac myosin heavy chain (*MYH7*) with missense mutations. A recent study showed that sarcomeric variants are mostly associated with female gender and young age, presenting asymmetric septal hypertrophy, a family history of HCM and SCD [107]. This pathology presents morphological and pathological heterogeneity, penetrance and age dependence with the consequence of different clinical outcomes, with conditions ranging from asymptomatic patients to cardiac arrhythmias and SCD [108].

Furthermore, it has been observed that genetic and environmental modifiers may also be involved in the pathogenesis of the disease. Other genes that can cause HCM include *TNNT2, TNNI3, TPM1, MYL2, MYL3* and *ACTC1* even if with lower frequency (1–5% each) [109–112]. In recent years, other genes besides sarcomeric ones have also been identified. These include *CSRP3, PLN, CRYAB, TNNC1, MYOZ2, ACTN2, ANKRD1 FLNC* [64,113–118] and *FHL1* [114,119].

Interesting results have also been obtained with the study of miRNAs, but this will be discussed later. Given the variety of genes involved, there is an equivalent high phenotypic variety of recognized HCM.

Unfortunately, the genotype–phenotype relationship may not be obvious. A high percentage of patients are asymptomatic or mildly asymptomatic, and the diagnosis is made during family screening or by accidental observation in mid or late adulthood where functional heart weakness has already progressed. Not surprisingly, members of the same family with the same mutation can have very different phenotypes [106]. Often, the phenotype is mild, but the arrhythmic prognosis could be particularly poor. Fortunately, the identification of causal mutations in a proband with HCM facilitates pre-symptomatic diagnosis of family members and clinical monitoring [97]. To explain the phenotypic variability, gene and environmental modulators were investigated. Interesting results have been obtained from miRNA studies which could therefore become potential biomarkers and therapeutic agents. If confirmed, the results obtained by the miRNAs would allow to have an early diagnosis, to evaluate the prognosis and future therapies. In this regard, the transcriptional profile modified by miRNA expression and other post-transcriptional modifications appears to be crucial for understanding the onset of HCM.

Currently, therapies for HCM are purely symptomatic. Beta-blockers, calcium channel blockers and disopyramide are generally given to patients [120]. Severe patients require surgical myectomy and alcohol ablation to relieve left ventricular obstruction [121].

MiRNA and Hypertrophic Cardiomyopathy

The discovery of miRNAs and their role at the post-transcriptional level has made it possible to identify pathological regulatory mechanisms that have been little known until now. To date, an ever-growing list of HCM-associated miRNAs varied in animal models and human samples has been created and the miRNA profile may allow understanding of the pathophysiology of the disease [122]. For this reason, some miRNAs have been reported as useful biomarkers of diseases and future therapeutic agents [82]. In this section, the best-known miRNAs involved in HCM will be evaluated (Table 2).

In one study, 370 miRNAs were analyzed in pathological HCM tissue, with and without mutation in the MYH7 gene. Two miRNAs—miR-590-5p and miR-92a—were overexpressed in HCM patients compared to healthy controls, and their expression was different between HCM-mutated and HCM-unmutated tissues [80]. In addition, the same group studied miRNA profile expression in plasma samples from HCM patients, where 10 miRNAs were analyzed in 24 patients and compared with healthy controls. Only miR-483-5p was upregulated in HCM patients compared to healthy ones [81].

Increased levels of miR-29a have been found in the blood of patients with HCM carrying mutations in the *MYH7* gene [82]. The same miRNA appears to be upregulated also in fibroblasts during the process of cardiac fibrosis. Therefore, miR-29a could be used as a biomarker of both fibrosis and hypertrophy [82]. Instead, miR-155 appears to be downregulated in HCM patients with *MYBPC3* mutations [84].

Duisters et al. have shown that in patients with LVH there is a correlation between the downregulation of miR-133 and the upregulation of CTGF and the effects of their interaction on collagen synthesis [83]; miR-133 is also downregulated in myocardiocytes during cardiac hypertrophy. RhoA (a GTP-GDP exchange protein), CDC42 (a kinase that transduces the signal involved in the development of hypertrophy) and WHSC2 (a nuclear factor involved in cardiogenesis) have been identified as possible target modulating negatively by miR-133 to counteract hypertrophy [83].

MiR-1 is downregulated in myocardiocytes in the process of cardiac hypertrophy. Under physiological conditions, this miRNA negatively modulates the Insulin-like Growth Factor-1 (IGF-1) pathway. In this way, it can block a series of processes including myocardial hypertrophy. Therefore, its downregulation can lead to cardiac hypertrophy [85].

C. Kuster and colleagues studied miRNA profile expression in six patients with a loss-of-function mutation in *MYBPC3* [86]. Among the 699 miRNAs analyzed, 13 of them

formed a unique miRNA signature for HCM: 10 were upregulated (miR-181-a2, miR-184, miR-497, miR-204, miR-222, miR-96, miR-34b, miR-383, miR-708 and miR-371-3p) and three were downregulated (miR-10b, miR-10a and miR-10b). Studies in silico demonstrated that a large number of differentially expressed miRNA-regulated genes were associated with the cardiac hypertrophic signaling pathway in which a large number of predicted mRNA targets were involved in β-adrenergic signaling [86]. In particular, they found that miR-204, incorporated in the *TRPM3* gene (Transient receptor potential cation channel subfamily M member 3), appears to be upregulated in HCM patients carrying a mutation in the *MYBPC3* gene, one of the genes most frequently mutated in HCM [86]. *TRPM3* encodes a cation-selective channel involved in calcium entry and is found to be upregulated with consequent alteration of calcium homeostasis in HCM [123]. This finding suggests that *TRPM3* may be involved in the pathogenesis of the disease caused by *MYBPC3* mutations.

MiR-139-5p is also one of the most downregulated miRNAs in the hearts of HCM patients [87,124].

Finally, Sun et al. screened miR-20, which was one of the highly expressed miR-NAs in HCM [88]. To do this, they constructed the cardiomyocyte hypertrophy model in vitro to validate whether miR-20 was associated with cardiomyocyte hypertrophy. In the study, a total of 1451 miRNAs were identified in both groups, 195 of which were upregulated and 172 downregulated in the HCM group compared to the control group; among these, miR-20a-5p was 2.26 times higher in the HCM group than in the control group. Over-expressed miR-20 could induce cardiomyocyte hypertrophy by suppressing *MFN2* expression. Among the target genes of miR-20 are *MFN2*, *PTEN*, *SMAD4* and *DUSP1*, which are associated with cardiac remodeling [88]. This study identified 367 miR-NAs differentially expressed between HCM cardiac samples and healthy control samples, meaning that some miRNAs may be indispensable in the development of HCM. It has been reported that miR-1 [125], miR-22 [126], miR-29a [82], miR-106a [127], miR-133 [128], miR-181a [129] and miR-195 [122] took part in the development of myocardial hypertrophy and showed significant differences between an expressed model of hypertrophic and physiological myocardium.

### 3.2. Dilated Cardiomyopathy

Dilated cardiomyopathy (DCM) is a major cause of SCD and heart failure with an unknown etiology. It has a prevalence of 1:2500, can occur at any age and is usually identified when associated with severe symptoms [130]. Unfortunately, it is the leading indication for heart transplantation in children and adults worldwide [98]. DCM is a heart muscle disease characterized by ventricular dilation and systolic dysfunction with cardiac fibrosis in the absence of abnormal load conditions or coronary artery disease [130,131]. It progressively leads to heart failure and a decline in left ventricular (LV) contractile function.

DCM is a complex disease with a common phenotype, but the pathological mechanisms are heterogeneous and still poorly understood. Early diagnosis is essential for the clinical management of the patient. Clinical diagnosis is currently made using imaging methods and genetic tests. However, the assessment of the disease still remains challenging; therefore, new non-invasive indicators are needed. The European Society of Cardiology has proposed a genetic and non-genetic classification of DCM, including etiologies such as peripartum, cancer therapies, drug or alcohol abuse, and myocarditis [132]. However, overlapping phenotypes often make the differential diagnosis unclear.

Currently, myocardial biopsies are used as a diagnostic tool. However, due to the invasiveness of the procedure, the use of the method is limited to a few patients [133]. The most used imaging techniques for the diagnosis of DCM are transthoracic echocardiography and magnetic resonance imaging as they are non-invasive and of wide applicability [134,135].

Approximately 20–35% of patients with DCM exhibit a family inheritance with incomplete penetrance associated with at least 40 genes [98]. Most DCM-associated mutations have been found in genes encoding proteins related to the cytoskeleton, sarcomere, nuclear envelope, ion channels and unclassified proteins [130,136]. Identifying the genetic factors

that lead to the clinical manifestation of diseases is the key to understanding the triggering mechanism that initiates the disorder [137]. In recent years, great efforts have been made to explore the molecular mechanisms underlying DCM which, however, still remain little known. Currently, attention is focusing on the myocardial gene expression of miRNA and their role in DCM [138,139]. Research suggests that miRNAs signature may constitute a novel source of non-invasive biomarkers for a wide range of cardiovascular diseases. Specifically, several studies have reported the potential role of miRNAs as clinical markers among the etiologies of DCM. However, this field has not yet been explored in detail.

MiRNA and Dilated Cardiomyopathy

Currently, several manuscripts have evaluated the relationship between miRNA and DCM profiles, without a detailed definition of the etiological mechanism. Among these is the manuscript of Tao et al., in which the interaction between long non-coding RNA (lncRNA), miRNA and competing for endogenous RNA (ceRNA) in patients with DCM were studied [140]. In this study, a miRNA array was initially performed to determine the differentially expressed miRNAs in samples from DCM patients and healthy controls. The results obtained were confirmed by RT-PCR. Cardiac tissue from patients with DCM and the mouse model of the pathology were used as samples. The results showed that miR-144-3p and miR-451a are downregulated compared to healthy controls, while miR-21-5p is upregulated [140]. Based on the ceRNA theory, a triple global network was developed using data from the NCBI-GEO (National Center for Biotechnology Information Gene Expression Omnibus) and the results obtained from the miRNA array. The results showed that two lncRNAs (NONHSAT001691 and NONHSAT006358) targeted at miR-144/451, both highly related to DCM. Therefore, clusterization and the use of an adequate random walk with restart algorithm for the analysis of the ceRNA network have identified four lncRNAs (NONHSAT026953/NONHSAT006250/NONHSAT133928/NONHSAT041662) that interact with miR-21 and are significantly related to DCM. This study could provide a new strategy for diagnosing DCM or other diseases. Furthermore, lncRNA–miRNA pairs can be considered potential diagnostic biomarkers or therapeutic targets for DCM [140].

In another study, 3100 miRNAs were evaluated in the plasma of four DCM patients. Forty-seven miRNAs were differentially expressed compared to three healthy subjects. Among these 47 miRNAs, miR-3135b, miR-3908 and miR-5571-5p were chosen because their levels were significantly increased in the plasma samples. The results obtained from this first comparison were then confirmed by a larger cohort, demonstrating that indeed the upregulation of miR-3135b, miR-3908 and miR-5571-5p had a discriminatory power to distinguish DCM patients from controls utilizing analysis of the ROC curve [89].

Highlighting the regulatory role that miRNAs can play, it has been understood that miR-148a is downregulated in DCM, while it is upregulated in concentric hypertrophy in human cardiac biopsies [90]. These results were confirmed in transgenic mouse models for DCM and concentric hypertrophy. The knockdown of miR-148a, obtained through the use of antagomiRNAs in WT mice, led to chamber dilation, increase in the volume of the left ventricle, cardiac wall thinning and reduction of the ejection fraction. Instead, the upregulation of AAV-mediated miR-148a protects against systolic dysfunction caused by pressure overload [90].

Yu et al., in 2011, evaluated the plasma levels of circulating miR-185 in patients with DCM [91]. These patients were compared with healthy ones. The analysis showed that miR-185 expression was significantly higher in DCM patients. DCM patients were placed in two distinct subsets, "high-group" and "low-group", clustered by miR-185 plasma levels. Essays were performed at DCM diagnosis time and 1-year follow-up. During this year, DCM cases had received standard therapy. At follow-up, circulatory levels of miR-185 were stable. Patients in the "high-group" showed evident improvements in left ventricular size and systolic function with a significant decrease in blood pressure and cardiovascular mortality. This suggests that higher levels of circulating miR-185 determine the better

clinical outcomes in DCM patients, proposing miR-185 as a new prognostic biomarker for DCM [91].

Interestingly, mutations in enzymes that participate in miRNA biogenesis can also induce DCM. Studies conducted on subjects with DCM have found altered levels of Dicer expression. Dicer targeted cardiac deletion was performed in a mouse model. The result was progressive DCM, heart failure and early postnatal lethality [141]. This led to a reduction in mature miRNA levels and an increase in pre-miRNA levels. Dicer expression was reduced in human patients with DCM or heart failure. However, therapeutic implantation of a left ventricular assist device resulted in an increased Dicer expression and improved cardiac function [141].

Despite all these data, further studies will have to be carried out to demonstrate a unique relationship between miRNA and cardiomyopathies in general; therefore, they can be used as early biomarkers of disease or for possible gene therapies.

### 3.3. Arrhythmogenic Cardiomyopathy

Arrhythmogenic cardiomyopathy (ACM) is a relatively rare genetic disease of the heart muscle with a frequency of 1:5000 [142–145], characterized by palpitations, syncope and/or cardiac arrest secondary to ventricular tachycardia (VT) or fibrillation; ventricular dysfunction and heart failure and high risk of SCD may also develop in some patients [146]. Structural changes include dilation of the right ventricle, aneurysms, abnormalities of regional wall movement, fibrosis, adipose infiltration and impaired ventricular function.

Histologically, cardiomyocyte death, inflammation and progressive substitution of adipose or fibro-adipose ventricular cardiomyocytes are observed [147–149]. Consequently, there is progressive atrophy of the ventricular myocardium which interferes with the conduction of the electrical impulse [98,150]. The disease progresses outwardly, first involving the sub-epicardial tissue and extending towards the endocardium, eventually resulting in a thinned and trans-mural lesion [150]. Inflammatory infiltrates are typical of the ACM and are often observed in both ventricular walls [151]. Generally, males are more frequently affected by ACM than females; therefore, it is thought that there is a correlation between the disease and sex hormones, especially on the severity of ACM [152]. ACM is a disease that exhibits variable expression and age-related reduced penetrance. Clinical symptoms typically present in the third to fourth decade of life, with arrhythmic manifestations that generally precede structural features. Sometimes ACM affects adolescents and rarely children [153]. The early onset of ACM can occur during adolescence and early adulthood and in many cases, they present with nonspecific symptoms, such as syncope and palpitations [154]. It particularly affects athletes, as intense exercise worsens the phenotype of the disease [154].

It is a cardiomyopathy caused by heterozygous mutations in genes coding mainly for proteins of the desmosomal protein complex, the adhesive junctions that connect cardiomyocytes, identified in almost 50% of subjects and with low and age-dependent penetrance [155]. However, the molecular mechanisms that lead to the destruction, remodeling and arrhythmic predisposition of the myocardium remain poorly understood. ACM inheritance is autosomal dominant with incomplete penetrance being the most common mode of transmission [156], although recessive forms are also known, namely, Naxos and Carvajal syndromes, and are associated with a cutaneous phenotype [157].

The most frequently mutated genes are desmosomal and cardiomyocyte junction genes [157], but mutations in non-desmosomal genes are also known [156]. Desmosomes influence intracellular transduction signals via the Wnt pathway, which is impaired in patients with ACM [92]. Cardiomyocytes form electrical and structural connections due to desmosomes, adherent junctions and gap junctions, located at the intercalated disc [158,159]. The most affected genes are on *JUP* and *DSP* [160–162], and truncating and missense mutations in the desmosome genes *PKP2* (encoding plakofilin 2), *DSG2* (encoding desmoglein 2) and *DSC2* (encoding desmocollin 2) in patients with ACM [163–165]. About half of ACM patients have mutations in one or more of these desmosomal genes [166,167]. *PKP2*

is the most commonly affected gene in adult cohorts [153,163], while some studies have suggested that the pediatric age group has more frequent mutations in the *DSP* [153,168].

Mutations have also recently been identified in the adherent junction genes *CDH2* (encoding cadherin 2) [169,170] and *CTNNA3* (encoding catenin-α3) [171] in patients with ACM.

Less frequent, there are also mutations in non-desmosomal genes. It is included genes that are involved in cytoskeletal architecture, calcium manipulation, sodium transport and cytokine signaling [172,173]. After desmosomal genes, genes encoding cytoskeletal proteins constitute the second largest category of ACM-associated mutations. These defects alter the architecture of cardiomyocytes [174,175]. The genes affected are *DES* (coding for desmin), *LMNA* (coding for lamina A), *TMEM43* (coding for transmembrane protein 43), *TTN* (coding for titin) and *FLNC* (coding for filamin C).

These pathways include canonical and non-canonical WNT signaling, the Hippo-Yes-associated protein (YAP) pathway and transforming growth factor-β signaling. Despite the discovery of multiple pathogenic genes, a large percentage of patients (35–50%) have no identifiable disease-associated variants. It suggests a more heterogeneous and complex etiology, with both polygenic and environmental factors contributing to the phenotypic expression [175,176]. For patients with an identifiable genetic cause, the exact biological mechanisms underlying this diverse and pleiotropic disease remain poorly characterized. Increasingly, asymptomatic relatives with variable penetration disease can be detected by cascade family screening [176].

Diagnosing ACM can be challenging and requires a high degree of clinical suspicion as well as supporting diagnostic tests. Primarily, it is based on a scoring system of criteria that include structural and electrocardiographic changes, tissue characterization, previous arrhythmic events and family history. In selected cases, a genetic test is recommended [168]. Criteria for the clinical diagnosis of ACM were defined by the International Task Force (ITF) to inform the diagnostic process and improve consistency between research studies [168]. These criteria consider cardiac morphology and function, tissue characterization, electrical rhythm conduction and family history, including identification of pathogenic mutations.

MiRNA and Arrhythmogenic Cardiomyopathy

The clinical heterogeneity of the ACM and its incomplete penetrance suggests that there are also other mechanisms, such as the involvement of more than one pathogenic allele and epigenetics [177,178]. These two factors contribute together with known gene mutations to the severity of the disease. Therefore, how the synergy of genetic, epigenetic and environmental factors acts to modify the phenotype and the onset of the disease is crucial for understanding the pathophysiology of the disease [179]. Unfortunately, few studies have evaluated the miRNAs circulating in the ACM.

Most of the miRNA identification studies were performed on serum or plasma samples, but also myocardial tissue samples from subjects with ACM. As previously reported, circulating miRNAs are extremely stable, often found in association with exosomes or proteins and represent potentially informative biomarkers [180]. The underexpression of specific miRNAs on tissues and the overexpression in the circulation lead us to hypothesize that miRNAs may be released into the circulation by apoptotic or necrotic cardiomyocytes. Consequently, elevated levels of a specific miRNA can indicate disease progression, as its level increases as more cardiomyocytes die. This evidence suggests that circulating overexpressed miRNAs may be potential prognostic biomarkers. Conversely, miRNA overexpression in tissue could indicate cardiomyocyte malfunction.

Zhang and colleagues studied miRNA profile expression in 24 histologically confirmed ACM patients compared with controls. They evaluated 1.078 miRNA levels by RT-PCR and identified 21miRNAs differentially expressed in ACM samples [93]. The data obtained were validated and the analysis of the ROC curve was performed to determine whether these miRNAs have diagnostic power. The conclusion was that the overexpression of miR-1251, miR-21-3p, miR-21-5p, miR-212-3p and miR-34a-5p and the downregulation of miR-135b,

miR-138-5p, miR-193-3p, miR-302b-3p, miR-302c-3p, miR-491-3p, miR-575, miR-4254 and miR-4643, allowed discrimination between ACM and healthy controls. In silico analyzes suggested the correlation of two miRNAs (miR-21-5p and miR-135) with the Wnt and Hippo pathways, which have been associated with the pathogenesis of ACM [92,181]. MiR-21-5p and miR-135 may play an important role in the regulation of Wnt/β-catenin and Hippo signaling pathways resulting in the phenotypic manifestation of ACM [93].

In 2017, Sommariva and colleagues identified miR-320a as a potential plasma biomarker of ACM. In particular, they correlated the ACM and low plasma levels of miR-320a [94]. In this study, 377 miRNAs present in the plasma of three ACM patients and three healthy controls were screened [94]. One-hundred-and-twenty-one miRNAs were detected in all plasma samples and five showed potential differential expression between ACM patients and controls. When these five miRNAs were evaluated in 36 ACM patients and 53 healthy controls, only miR-320a exhibited significantly lower expression. Plasma levels of miR-320a showed a 0.53-fold difference in expression between ACM and healthy control. Furthermore, the miRNA expression profile in ACM patients was also compared to patients with idiopathic ventricular tachycardia (IVT) and miR-320a showed 0.78-fold lower expression, suggesting not only miR-320a as a potential biomarker for patients with ACM, but also as a potential discriminatory biomarker for ACM vs. IVT. Plasma levels of miR-320a were not influenced by intense physical activity, so the authors hypothesized that the differential expression of miR-320a is independent of the increase in mechanical elongation and adaptive cardiac remodeling induced by the training [94]. Furthermore, miR-320a appeared to have mechanistic implications in the pathogenesis of ACM. The diagnostic value of miR-320a must be validated in larger cohorts of patients with ACM.

A similar study was performed on 62 patients with ventricular arrhythmia (VA), where 28 had definite ACM, 11 had borderline ACM and 23 had IVT [95]. In this study, they observed plasma levels of miR-144-3p, miR-145-5p, miR-185-5p and miR-494 with significantly higher expression in ACM patients with VA than in healthy controls. Among these, miR-494 levels appeared to have a central prognostic value because it was linked to recurrent VA after ablation [96]. Furthermore, based on in vitro results, a possible correlation between high expression of miR-494 and the apoptotic process that occurs in ACM hearts, although the role of miR-494 in apoptosis is unclear [96].

## 4. How to Train a miRNA: A Possible Therapy

Depending on how miRNAs are dysregulated when the heart is under stress, their manipulation has great potential for developing new treatments to restore the normal phenotype.

The central action of miRNAs is to suppress protein expression by binding and silencing specific target mRNAs, which reduce protein synthesis. The overexpression of a miRNA will suppress its direct targets, while the inhibition of endogenous miRNAs will decrease the expression and therefore gene inhibition is lacking. Therefore, miRNAs constitute extremely attractive targets for possible therapies [182,183].

The strong impact of miRNAs on the cardiac phenotype is of particular interest in the possibility of targeting these molecules as therapeutic substrates.

The most effective way to use miRNAs as drugs is to modulate intracellular levels of miRNAs by transfecting target cells with miRNA mimics or inhibitors (Table 3). However, these methods require the development of efficient methods for cell/site-specific release, which we have not been able to achieve for the moment. As reported in the introduction of this review, a single miRNA can affect several genes at the same time. This offers miRNAs both an advantage and a disadvantage over conventional drug therapies, which traditionally target a single target within a cellular pathway [184]. To date, few clinical trials are in Phase I that show significant clinical promise of miRNAs. Only one treatment for chronic hepatitis C has managed to move into phase II clinical trials.

**Table 3.** The potential advantages and disadvantages of using miRNAs as an alternative therapy to conventional drugs.

| miRNA Therapeutic Approach | Advantage | Limitation |
|---|---|---|
| miRNA mimics | Promote the expression of miRNAs | Low efficiency in the heart and vascular system; Can cause miRNA to over-act, potentially causing serious side effects; Easily degraded by nucleases; The chemistry of the construct is toxic; |
| miRNA inhibitors | Block the activity of miRNAs Directly bind the target of the miRNA sequence | Low efficiency in the heart and vascular system; Low target binding affinity; Unwanted genetic changes or off-target effects; Easily degraded by nucleases Difficult to create and to keep stable |

Currently, the activity of miRNAs can be modulated by different approaches based on the imitation of the functions of miRNAs or the silencing of their function due to the use of miRNA inhibitors (antagomiRNAs and miRNAsponges) (Figure 3) [7].

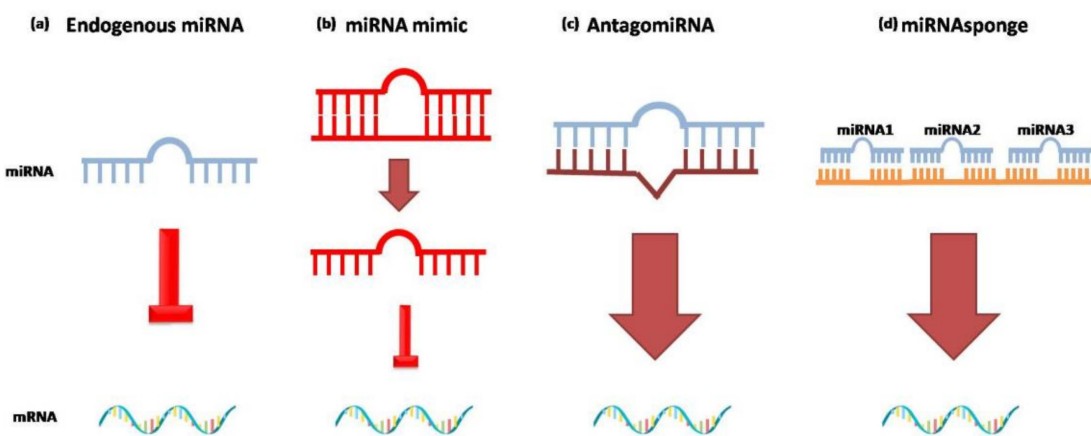

**Figure 3.** Different approaches to using exogenous miRNAs as gene therapies. (**a**) Endogenous miRNA (blue) binds to the seed sequence present in the 3′-UTR of the mRNA target; (**b**) representation of miRNA mimic (red), a synthetic double-stranded RNA molecule. The double-stranded structure mimics the endogenous miRNA*duplex that binds RISC to inhibit translation of the mRNA target. (**c**) AntagomiRNA (brown) synthetic miRNA complementary to the target miRNA that has to be inhibited. (**d**) miRNAsponge(orange) synthetic oligonucleotide containing several seed sequences for different multiple miRNAs.

If miRNAs levels are compromised in any way, exogenous miRNAs can be administered in vivo. MiRNAmimics are small chemically synthesized double-stranded RNAs that mimic endogenous miRNAs and cause gene silencing. One strand of this molecule is identical to the native form of the miRNA, the other is complementary. The double-stranded structure is needed so the RISC complex can accurately recognize miRNAmimics [185].

AntagomiRNAs are synthetic single-stranded RNAs made up of nucleotides complementary to an endogenous miRNA so that they can be silenced. The inhibitors are complementary to the entire sequence or the seed sequence of the miRNA [185]. This method allows the repression of an over-regulated miRNA by blocking its inhibiting action [186].

MiRNAsponges are another approach to reduce miRNA levels. MiRNAsponges are exogenous transcripts that contain complementary regions for multiple miRNAs that have the same target site. The release of miRNAsponges into a cell determines their binding to the target miRNAs and reduces the number of free and active miRNAs [185].

To enter the cell, these external miRNAs have to pass the lipid bilayer of the cell membrane. The lipid bilayers allow small neutral and slightly hydrophobic molecules to passively diffuse through them, while preventing large, charged molecules, such as RNA, from passing through them [187]. To do this, an approach has been developed that exploits the use of chemically modified liposomes or cationic polymers with specific ligands for the receptors on target tissues to improve the ability of nanoparticles to bypass the plasma membrane and enter target cells [188]. These modifications allow cellular uptake of exogenous miRNAs via receptor-mediated endocytosis.

Alternatively, other systems exploit recombinant viral systems such as lentiviruses, adenoviruses and adeno-associated viruses (AAVs) as vectors.

Currently, viral vectors are best suited for delivery miRNAs in the myocardium [189]. Viruses have already been used for some time to target genetic material in a given cell. In recent studies, excellent results have been obtained at the cardiac level through the use of lentiviruses and AAVs.

In one experiment, miR-378, a regulator of cardiac hypertrophy, was administered via AAV9 (a serotype with cardiac tropism) in vivo, improving cardiac function [190]. AAV9 administration has also been exploited to treat dilated cardiomyopathy (DCM) in mouse models. The resulting is overexpression of miR-669a, downregulated in DCM. This increase in miR-669a levels reduced cardiac fibrosis, hypertrophy and apoptosis of cardiomyocytes for up to 18 months [191].

Currently, AAV cardiotropic viruses achieve efficient miRNA delivery in cardiomyocytes [190]. However, this mechanism can only be used to inhibit overexpressed miRNAs. The use of antagomiRNAs or miRNAsponges are not suitable methods for the overexpression of an endogenous miRNA. However, antagomiRNAs have been observed to have low efficiency in the heart and vascular system. Therefore, the use of exogenous miRNAs for cardiovascular applications will require solutions for local or specific cell type release.

Another method to allow the entry of miRNAs into cardiomyocytes is ultrasound-mediated sonoporation [192]. It uses albumin-coated microbubbles, which carry genetic material to target sites. Microbubbles are gas-filled acoustic microspheres that explode with ultrasound and release their contents to the target site [193]. The ideal would be to develop new techniques that use electromechanical mapping [194] or the use of positron emission tomography to study blood flow to carry miRNAs into cardiomyocytes [195].

Unlike conventional drugs, which are specific for a cellular target, a single miRNA does not act on just one target, but on multiple ones. The multiple mechanisms of action of miRNAs can cause numerous side effects if they are released into the bloodstream. Potentially, most miRNA-based therapies would act systemically, which could preclude widespread clinical use. The development of insertion method of highly site-specific miRNA could be a step forward for effective clinical use, leading to the development of miRNA-based therapies [196].

In the future, it is hoped that exploiting miRNAs will provide an effective therapeutic tool in the field of vascular and cardiovascular biology. Monogenic therapy has had limited success, and a single miRNA has far greater therapeutic potential with its unique ability to alter complex genetic networks. Therefore, it is hoped that miRNAs will become a new line of treatment in multiple human diseases. However, at the moment, the limitations are greater than the potential therapeutic benefits.

## 5. Conclusions

The discovery of miRNA changed our understanding of gene expression regulation. Indeed, miRNAs can regulate the expression of proteins at the post-transcriptional level and are involved in cardiovascular physiology, while their expression is altered in various cardiovascular diseases. Many different biological events interact to determine the cardiovascular phenotype and its response to injury or stress. Due to their multitarget ability, a multitude of miRNAs is involved in these processes. Studies on miRNAs in cardiomyopathy could represent an important advance both for their use as biomarkers

and for their therapeutic potential [7]. Treatment strategies currently focus on the systemic delivery of exogenous miRNAs. Exogenous miRNAs act by miming the function of endogenous ones, while exogenous complementary sequences bind miRNAs and limit their role, but every miRNA act on multiple genes. Systemically delivered drugs reach virtually every body district, acting also in non-desired organs and potentially leading to side effects and iatrogenic pathology. Due to their lack of targeting, future efforts should be aimed at evaluating site-specific strategies. For the cardiovascular system, targeting could be achieved by the use of adeno-associated viruses as vectors for the release of miRNAs or antagomiRNAs linked to nanoparticles or miRNA mimics [197].

In hereditary cardiomyopathies, miRNAs could provide an answer to the search for factors responsible for broadly variable expressivity and thus bridge the genotype–phenotype gap, improving the therapeutic strategy. Research has mainly focused on identifying the mechanisms within a single tissue or cell type. However, care must be taken because the regulation of miRNA protein expression is highly dependent on the context and cell type. Therefore, their ubiquitous expression makes the side effects of miRNA therapies unpredictable. Targeting individual miRNAs, therefore, requires a meticulous evaluation of systemic effects.

Although several studies have identified several altered circulating miRNAs in the plasma or serum of patients with cardiomyopathy, these are differentially expressed across disease phenotypes and are potentially usable as a novel early non-invasive biomarkers. To achieve this, however, more in-depth studies and standard protocols on the mechanics of these non-coding RNAs and their validation in larger cohorts of patients are needed to use them as disease biomarkers. In the same way, it is essential to develop faster analytical technologies, to make the use of miRNAs as biomarkers effectively competitive with, respect to current techniques.

**Funding:** This research received no external funding.

**Conflicts of Interest:** The authors declare no conflict of interest.

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
