# Peer review of "MicroRNAs: From Junk RNA to Life Regulators and Their Role in Cardiovascular Disease"

_cardiogenetics, doi:10.3390/cardiogenetics11040023_

Round 1

Reviewer 1 Report

In the paper, titled "MicroRNAs: From junk RNA to life regulators and their role in cardiovascular disease", Amodio F et al., summarize the current literature regarding the role of miRNAs in regulating cardiovascular diseases. Overall, the manuscript provides an interesting insight into miRNA biology and it impact on progression of cardiac disease. There are some structural issues that need to be addressed and will provide clarity to the subject matter.

  1. It would be more beneficial for the readers to include gene targets of the miRNAs is Table 2.
  2. Also, please include a separate table that shows cardiac cell (cardiomyocyte, endothelial cells and fibroblast) specific miRNAs, their gene targets and potential outcomes. 
  3. In the abstract, authors mention use of miRNAs as biomarkers for CVD including their release in the circulation. However, this subject is not discussed in the manuscript.
  4. It would be nice to include a table highlighting advantages/limitations of miRNA-based therapeutic approaches.
  5. Figure 2, typo for "translation". Please correct. Also, the manuscript needs a thorough read as there are multiple errors of grammar and syntax.

Author Response

Point 1: It would be more beneficial for the readers to include gene targets of the miRNAs is Table 2.

Response 1: We have added a new column to Table 2. Since there are few studies related to miRNA and cardiomyopathies (and that the same miRNA has multiple target genes), we searched in specific databases for miRNAs target genes and verified an association between the genes and diseases.

Point 2: Also, please include a separate table that shows cardiac cell (cardiomyocyte, endothelial cells and fibroblast) specific miRNAs, their gene targets and potential outcomes. 

Response 2: We would like to satisfy your request, but unfortunately, we do not yet have this type of information since there are still no studies that analyze on which specific cells certain miRNAs act and, consequently, the potential outcomes. Therefore, it is not possible to fulfill this request.

Point 3: In the abstract, authors mention use of miRNAs as biomarkers for CVD including their release in the circulation. However, this subject is not discussed in the manuscript.

Response 3: Thank you for your comment, we have added the following sentence:

“Previously, we indicated that mature miRNAs, pre-miRNAs and stand passengers can be found in the blood due to their inclusion in microvesicles. Therefore, it is possible to measure these molecules not only in the tissues, but also in the blood of patients with CMP.”

Point 4: It would be nice to include a table highlighting advantages/limitations of miRNA-based therapeutic approaches.

Response 4: Thank you for your precious comment. We have added table 3 as required.

Point 5: Figure 2, typo for "translation". Please correct.

Response 5: Thank you. We corrected the spelling error.

Point 6:Also, the manuscript needs a thorough read as there are multiple errors of grammar and syntax.

Response 6: Thanks a lot. We checked the grammar and syntax.

Reviewer 2 Report

The manuscript is generally well written and it covers a very important topic. There are few minor comments:

In the abstract the authors stated: “More than 1,000 miRNAs are encoded in the human genome (approximately 5% of the genome)”, while in the introduction, they stated: “To date, approximately 2.500 miRNAs have been identified in the human genome. MiRNAs are essential in various biological processes, including cell differentiation and proliferation, cell death and metabolism ”. Later on, the authors quote 3,100 miRNA. Although, not technically erroneous statements, but a reconciliation between the numbers appear in order.

In the sentence: “With NGS technologies it is possible to obtain a rapid and accurate measurement of miRNAs which can also be measured also with real-time PCR (even in small volumes of biofluids)”, please elide one “also”.

Also, in the phrase: “Furthermore, more miRNAs can regulate more genes and at the same time more genes can be regulated by more miRNAs, so there is a low correlation between miRNA and specific pathologies”, please use a synonym for “more’ so you do not have to repeat it.

The sentence: “Cardiomyopathies are a very heterogeneous group of diseases affecting the heart muscle characterized by changes in the size of cardiac chambers, ventricular wall thick- ness, or abnormal” appears unfinished.

Please, check English grammar

Author Response

Point 1: In the abstract the authors stated: “More than 1,000 miRNAs are encoded in the human genome (approximately 5% of the genome)”, while in the introduction, they stated: “To date, approximately 2.500 miRNAs have been identified in the human genome. MiRNAs are essential in various biological processes, including cell differentiation and proliferation, cell death and metabolism”. Later on, the authors quote 3,100 miRNA. Although, not technically erroneous statements, but a reconciliation between the numbers appear in order.

Response 1: Thank you for pointing out this numerical difference. We modified the abstract as follows and removed the second statement you indicated.

“About 5% of the genome encodes miRNAs which are responsible for regulating numerous signaling pathways, cellular processes and cell-to-cell communication.”

Point 2: In the sentence: “With NGS technologies it is possible to obtain a rapid and accurate measurement of miRNAs which can also be measured also with real-time PCR (even in small volumes of biofluids)”, please elide one “also”.

Response 2: Thank you. Done

Point 3: Also, in the phrase: “Furthermore, more miRNAs can regulate more genes and at the same time more genes can be regulated by more miRNAs, so there is a low correlation between miRNA and specific pathologies”, please use a synonym for “more’ so you do not have to repeat it.

Response 3: Thank you. We have rewritten the sentence as follows:

“Furthermore, different miRNAs can regulate more genes and at the same time,  several genes can be regulated by more miRNAs, so there is a low correlation between miRNA and specific pathologies.”

Point 4: The sentence: “Cardiomyopathies are a heterogeneous group of diseases affecting the heart muscle characterized by changes in the size of cardiac chambers, ventricular wall thickness, or abnormal” appears unfinished.

Response 4: Thank you for your comment. We corrected the sentence as follows:

“Cardiomyopathies are a very heterogeneous group of diseases that affect the heart muscle. They are generally characterized by changes in the size of the heart chambers, ventricular wall thickness, or contraction abnormalities.”

Point 5: Please, check English grammar

Response 5: Thanks a lot. We checked the grammar and syntax.

Round 2

Reviewer 1 Report

There are no further comments